

# The flask monitoring program for high-precision atmospheric measurements of greenhouse gases, stable isotopes, and radiocarbon in the central Amazon region

Carlos A. Sierra[1], Ingrid Chanca[1,8,12], Meinrat Andreae[10], Alessandro Carioca de Araújo[2], Hella van Asperen[1], Lars Borchardt[1,9], Santiago Botía[1], Luiz Antonio Candido[3], Caio S.C. Correa[4], Cléo Quaresma Dias-Junior[11], Markus Eritt[1,9], Annica Fröhlich[1,9], Luciana V. Gatti[4], Marcus Guderle[1], Samuel Hammer[5], Martin Heimann[1], Viviana Horna[1], Armin Jordan[1], Steffen Knabe[1], Richard Kneißl[1,9], Jost Valentin Lavric[1,7], Ingeborg Levin[5, †], Kita Macario[6, 12], Juliana Menger[1], Heiko Moossen[1], Carlos Alberto Quesada[3], Michael Rothe[1], Christian Rödenbeck[1], Yago Santos[3], Axel Steinhof[1], Bruno Takeshi[3], Susan Trumbore[1], and Sönke Zaehle[1]

[1]Max Planck Institute for Biogeochemistry, Jena, Germany
[2]Empresa Brasileira de Pesquisa Agropecuária, Belém, Brazil
[3]Instituto Nacional de Pesquisas da Amazônia, Manaus, Brazil
[4]National Institute for Space Research, São José dos Campos, Brazil
[5]Heidelberg University, Institut für Umweltphysik, Heidelberg, Germany
[6]Universidade Federal Fluminense, Programa de Pós-graduação em Geociências (Geoquímica), Niterói, Brazil
[7]Acoem GmbH, Hallbergmoos, Germany
[8]Laboratoire des Sciences du Climat et de l'Environnement, Gif-sur-Yvette, France
[9]ICOS Flask- und Kalibrierlabor, Jena, Germany
[10]Max Planck Institute for Chemistry, Mainz, Germany
[11]Instituto Federal de Educação Ciência e Tecnologia do Pará, Campus Belém, Belem, Brazil
[12]Universidade Federal Fluminense, Instituto de Física, Laboratório de Radiocarbono, Niterói, Brazil
[†]deceased, 10 February 2024

**Correspondence:** Carlos A. Sierra (csierra@bgc-jena.mpg.de)

**Abstract.** Long-term and high-precision measurements of the mole fraction of greenhouse gases (GHG), together with their isotopic composition, are of fundamental importance to understand land-atmosphere interactions. Current flask monitoring programs have important information gaps in large regions of the Earth, particularly in the southern hemisphere and in continental tropical regions. Here, we report on the initiation of a monitoring program and the resulting dataset of high-precision GHG
measurements at the Amazon Tall Tower Observatory (ATTO), located in the central Amazon region of Brazil. In September 2021, we installed an automated flask sampler designed and built by the Integrated Carbon Observation System (ICOS) to collect air samples in 3-liter flasks at a height of 324 m above ground level. Samples are collected weekly, during a one-hour integration time between 13:00 and 14:00 h local time (17:00-18:00 UTC). The flasks are shipped to Jena, Germany, for analyses of $CO_2$, $CO$, $CH_4$, $N_2O$, $H_2$, $SF_6$, $^{13}C-CO_2$, $^{14}C-CO_2$, $^{18}O-CO_2$, $^{13}C-CH_4$, $^{2}H-CH_4$, $O_2/N_2$, and $Ar/N_2$ at the laboratories
of the Max Planck Institute for Biogeochemistry (MPI-BGC). Measurements from this monitoring program provide reference information for this site and act as an additional independent quality control for other measurements in the region. The record of $SF_6$ and simulations based on a regional atmospheric transport model suggest that the footprint of the measurements is





predominantly from the southeasterly and northeasterly directions. The time series of the different gas species measured in this monitoring program are being made publicly available through the ATTO data portal and the atmospheric flask sampling
program of the MPI-BGC.

## 1  Introduction

The Amazon river basin is one of the largest forested regions on Earth and exchanges large amounts of energy, water, and greenhouse gases (GHGs) with the atmosphere. Given the large area of the Amazon forest, gas exchange between the forest and the atmosphere has a considerable impact on atmospheric concentrations of GHGs at regional and global levels (Artaxo
et al., 2022; Gatti et al., 2021; Basso et al., 2023). Gas exchange between the forest and the atmosphere also carries a signal of metabolic activity and ecosystem functioning. For instance, Amazon forests release large quantities of carbon dioxide and methane to the atmosphere through the combined activity of autotrophic and heterotrophic organisms, which modify the isotopic signature of these gases through fractionation processes, providing a signature of their metabolism and the source of emissions (Ometto et al., 2002).

Previous studies based on atmospheric profiles of GHGs at the basin scale have shown signals not only on the carbon source/sink status of Amazon forests, but also on the signatures of fires and deforestation related to droughts (Andreae et al., 1988; Lloyd et al., 2007; Andreae et al., 2012; Gatti et al., 2014; van der Laan-Luijkx et al., 2015; Alden et al., 2016; Gatti et al., 2021; Basso et al., 2023). Measurements of methane from these aircraft campaigns have also shown the importance of biogenic fluxes at the basin scale and how climate variability influence methane fluxes for different regions of the Amazon
basin (Beck et al., 2012, 2013; Basso et al., 2021). These high-precision measurements of GHG collected in flasks from aircraft campaigns have demonstrated the value of monitoring the temporal evolution of GHG concentrations for the Amazon forests, a region where very few continuous monitoring sites exist (c.f. Molina et al., 2015; Botía et al., 2024).

Despite these previous efforts, the Amazon basin remains underrepresented in global observation networks of GHGs and contributes a large share of uncertainty in flux estimates from global atmospheric inversions. Furthermore, continuous records
of isotopes in gases are practically inexistent for the Amazon and most parts of the tropics, which hinders our ability to disentangle the main processes that contribute to GHG emissions from these regions.

In particular, radiocarbon measurements in carbon dioxide ($^{14}C - CO_2$) provide key information to disentangle the contribution of fossil fuel burning from the contribution of biogenic fluxes to the atmospheric $CO_2$ concentration record (Turnbull et al., 2009; Graven et al., 2020; Levin et al., 2022). Despite their importance, continuous high-precision measurements of $^{14}C - CO_2$
are only available at a handful of stations (Levin et al., 2022), and direct atmospheric measurements for the Amazon basin have never been done before on a regular basis. One of the main limitations for these measurements is the collection of large volumes of air to extract enough carbon in $CO_2$ for measurements by accelerator mass spectrometry, a technique that requires sophisticated laboratories, and therefore challenging logistics to frequently transport samples between remote field sites in the Amazon forest and laboratories in other countries or continents.





High-precision analysis of a comprehensive range of gas species and their isotopic signatures within the same sample, including radiocarbon, requires the collection of relatively large air volumes, with a minimum of 2 liters for natural atmospheric concentrations. Sampling at regular frequencies is now facilitated by new instrumentation that allows one to automatically program sampling events for the collection of air samples in glass flasks following standardized protocols for the timing of sampling, the flow rate of air for sample collection, the drying of air before storage in flasks, and the pressure at which the air

is stored (Levin et al., 2020). The Integrated Carbon Observation System (ICOS), a large European research infrastructure for monitoring GHGs, has developed such an automated sampling system, which can be deployed to remote sites with a minimum infrastructure to access relevant heights in the lower troposphere to collect representative samples of air. Such infrastructure is provided in the Amazon basin by the Amazon Tall Tower Observatory (ATTO) (Andreae et al., 2015), a research site in the central Amazon region with a tall tower of 325 m height above ground level (agl), which provides a unique opportunity to

collect gas samples at unprecedented heights for the region. In this article, we report on the initiation of a continuous monitoring program for gas sampling at the tall tower at the ATTO site for the continuous high-precision measurements of a set of GHGs and isotopes.

    The flask monitoring program at ATTO has four main objectives: (i) to provide a record of weekly measurements of $CO_2$, CO, $CH_4$, $N_2O$, $H_2$, $SF_6$, $\delta^{13}C-CO_2$, $^{14}C-CO_2$, $\delta^{18}O-CO_2$, $\delta^{13}C-CH_4$, $\delta^2H-CH_4$, $O_2/N_2$, and $Ar/N_2$ from samples taken at a

height of 324 m (agl) representative of background air for the study site; (ii) to provide additional independent quality control for other high-frequency measurements of $CO_2$, $CH_4$, CO, $N_2O$, and $\delta^{13}C-CO_2$ performed with other instruments at the site; (iii) to provide information on the background concentration of radiocarbon in $CO_2$ and other isotopes necessary for source partitioning of atmospheric signals at the local level; (iv) to detect anthropogenic signals related to large-scale fires and fossil fuel burning using a combination of tracers such $CO_2$, CO, and $\Delta^{14}C-CO_2$.

This article introduces the set of sampling and laboratory methods used at the ATTO flask monitoring program as well as data processing workflows. The data produced in this monitoring program is made available through the ATTO data portal (https://www.attodata.org). This article provides the main reference for the data and will be updated on a regular basis.

## 2    Methods

### 2.1    Study area and sample collection

The Amazon Tall Tower Observatory (ATTO) is a research infrastructure located in the central Amazon region of Brazil, and it is part of the Uatumã sustainable development reserve. The site is located 150 km from the city of Manaus in the northeast direction (02° 08.7520′ S, 59° 00.3350′ W) on top of a plateau at 130-140 m above sea level (asl). Vegetation in the region is characteristic of the tropical rain forest biome, and several ecosystems are found in the surroundings of the ATTO site, among them seasonally flooded black-water forests (*igapós*), white-sand forests (*campinas* and *campinaranas*), and non-

flooded evergreen rainforest (*terra-firme*). Surrounding the tower, vegetation is a typical *terra-firme* forest with an average tree height of ∼30 m and aboveground biomass carbon of ∼170 Mg C ha$^{-1}$ (Andreae et al., 2015). Mean annual precipitation is about 2382 mm, with a marked seasonal cycle of precipitation, including a rainy season between the months of February to



May, and a dry season from June to October (Botía et al., 2022). Mean annual temperature is 26.3°C, with a mean temperature of 27.5°C for the dry season and a mean temperature for the rainy season months of 25.2°C (Schmitt et al., 2023; Gonçalves et al., 2024). The site hosts three towers for the study of forest-atmosphere interactions, two towers of ∼ 80 m height with instrumentation for continuous measurements of GHG and aerosols, and the tall tower of 325 m height agl (331 m including the top lighting rod antenna). The flask monitoring system described in this article is installed on the tall tower.

The inlet air filter for sampling (Solberg F-15-050, Canada) is located on the top of the tall tower at a height of 324 m agl. At the time of installation, the estimated height of the inlet was 321 m agl, and previous studies commonly report this height, but a recent reassessment provided an actual height of 324 m agl. The air inlet is attached to a fluoropolymer (PFA) tubing of 6.35 mm (1/4 inch) outer diameter and ∼4 mm (5/32 inch) inner diameter (TILM07B, SMC Corporation, Japan). The tubing enters a temperature-controlled container at the base of the tower, where the automated flask sampler is located.

## 2.2 Automated flask sampler

In September 2021, we installed an automated flask sampler, designed and built by the ICOS Flask and Calibration Laboratory (FCL) (Levin et al., 2020). The sampler is a system that consists of four trays with a total capacity for 24 flasks, and an air drying unit through which the air passes before entering the flasks (Figure 1). The sampler has its own software for programming sampling events and can be controlled remotely. Similar samplers have been installed in all Class 1 stations of the ICOS network in Europe (∼ 19 sites as of January 2025) (Levin et al., 2020).

The glass flasks are of 3 liter volume (ICOS-3000, Pfaudler Normag Systems GmbH, Germany) and are covered by a dark plastic material that serves for mechanical protection and to prevent photochemical changes to the air samples. To enable flushing of the flask before air sampling, each flask has one valve at each end, which are connected to the sampler with a 12.6 mm clamp-ring connector. The valves at the ends of the flasks are made from polychlorotrifluoroethylene (PCTFE) sealing caps, which minimizes storage effects on trace gas composition in comparison to PFA seals (Rothe et al., 2005).

The air dryer contains two alternately operating water traps for permanent air drying down to a dew point of < -38°C. Each trap consists of a Dewar vessel with two glass cylinders (50 and 70 mm in diameter), two temperature sensors, a heating cartridge and a cooling probe that is connected to an immersion cooler. The cooling medium in the Dewar vessels is silicon oil M5 (Carl Roth GmbH & Co. KG, Germany). Inside the device are two immersion coolers, a valve installation with humidity sensors, an over-pressure pump for water removal from the heated traps, and the control electronics.

## 2.3 Flask sampling and measurement cycle

### 2.3.1 Sample collection and transport

Flasks are first prepared at the ICOS FCL in Jena, Germany, where they are inspected and tested for leaks. Each flask is filled with dry ambient air of known composition at a pressure of 1.6 bar. Sets of 12 flasks are packed in boxes with protective material and sent by air freight to the National Institute for Amazon Research (INPA) in Manaus, Brazil (Figure 2). From INPA, they are sent to the ATTO research station on a 4-6 hour trip that involves transport by car until Porto Morena, near





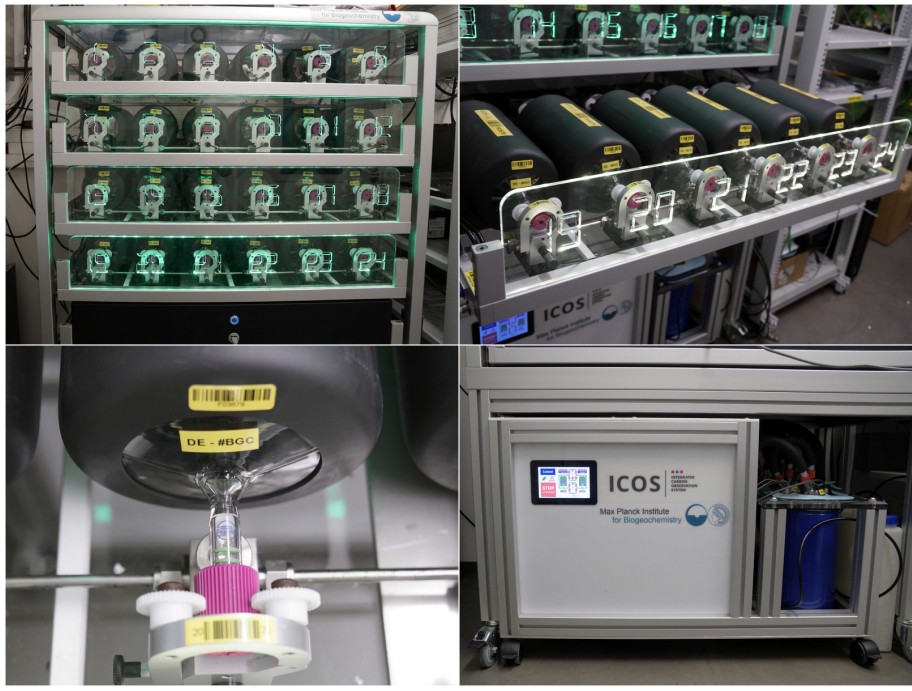

**Figure 1.** Automated flask sampler installed at the ATTO site. The sampler consists of four trays, and each tray holds six flasks, for a total of 24 flasks available for sampling. A drawer below the last tray contains all the electronic components and the compressor pump. The flasks are of 3 liter volume and samples are stored at an overpressure of ∼0.7 bar. A drying unit (bottom right) takes the air drawn in by the sampler and cools it to a dew point below -38°C.

the Balbina hydroelectric dam, transport by boat along the Uatumã river, and transport by car from the the river to the ATTO station.

The flasks are then loaded in the auto sampler, which is programmed to collect one sample every Thursday at mid-day local time. Sampling starts at 16:30 UTC (12:30 local time) with an initial 30 min flush of the sampling line. The collection of gas in the flasks starts at 17:00 UTC and follows the so-called $1/t$ method, which is a dynamic filling method with a maximum flow

rate of 2 L min$^{-1}$ at the beginning of sampling and a minimum of 80 mL min$^{-1}$ at the end of sampling. Filling of gas in the flask ends at 18:00 UTC. During a phase test that lasted about 4 months, we also collected samples at local midnight, between 05:30 and 07:00 UTC, to test the range in values obtained at the site.

Once two or more sets of 12 flask samples (12 flasks per shipping box) are collected, they start their return to Germany for laboratory analyses. They are transported back by boat and car to INPA in Manaus, and from there by air freight to Germany.

When the samples arrive at the MPI-BGC in Jena, they are assigned a Unique Sample Number (USN) and follow standard protocols for gas measurements at the gas, stable isotope, and radiocarbon laboratories. After measurements, the flasks are sent back to ICOS FCL where they are inspected and prepared, and the cycle starts again (Figure 2).



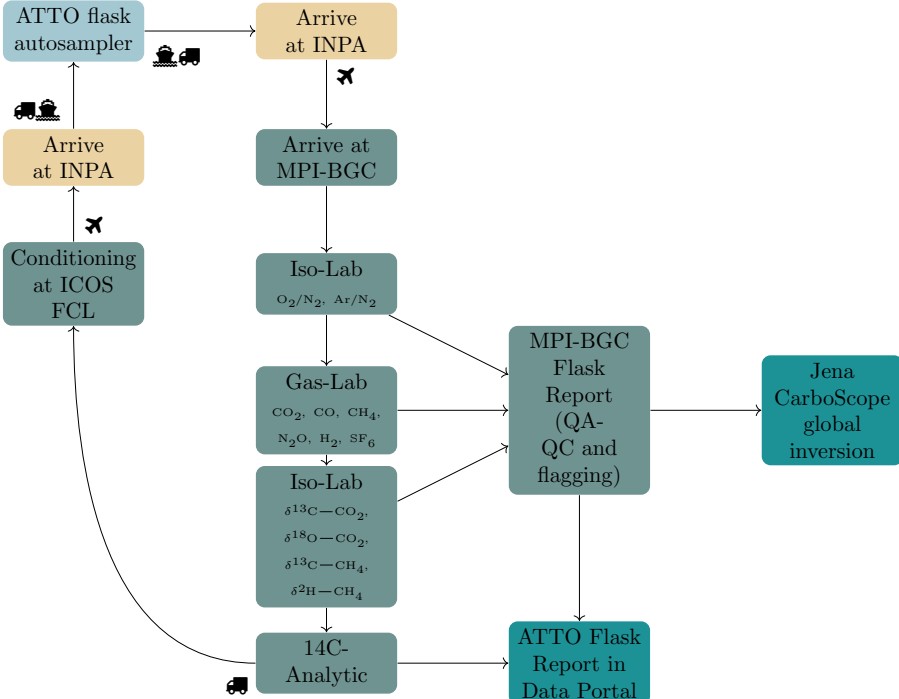

**Figure 2.** Flask sampling and analysis cycle for the ATTO flask monitoring program. The cycle starts with the preparation and conditioning of flasks at the ICOS-FCL, which are then transported to Brazil for air sampling at the ATTO tower (upper left corner). After sampling, sets of 12 flasks are sent from ATTO to the National Institute for Amazonian Research (INPA) in Manaus. From there, flasks are sent to the Max Planck Institute for Biogeochemistry in Jena, Germany. After arrival, the flasks are analyzed at the Iso-Lab and the Gas-Lab. These labs produce a report that feeds to the global network of monitoring sites of MPI-BGC (Heimann et al., 2022). This data report is also used to update the Jena CarboScope atmospheric inversion system. The remaining gas in the flasks is transferred to the 14C-Analytic lab for analysis of $^{14}C-CO_2$. Empty flasks are returned to the ICOS FCL for conditioning and the cycle starts again.

### 2.3.2 Description of laboratory analyses

After flasks arrive at the MPI-BGC and are assigned a USN, they pass through a sequence of analytical measurements by gas
chromatography, mass spectrometry, and accelerator mass spectrometry. The flasks follow the same analytical methods as all other flasks from the MPI-BGC flask network (Heimann et al., 2022), with the main difference that for ATTO, we do not collect replicates due to the design of the ICOS autosampler (Levin et al., 2020), and we add the radiocarbon measurements.

In a first step, samples are analyzed for $O_2/N_2$ and $Ar/N_2$ ratios. Flasks are attached to a customized autosampler and the air is then analyzed using an isotope-ratio mass spectrometer (IRMS, Delta+XL, Thermo Finnigan) via a dual inlet setup. Mea-
surements are made against reference air prepared in-house, which is periodically compared to reference air from the Scripps Oceanography Institution. Efforts are underway to bring measurements onto the new "SIO 2017 O2/N2" scale. Additional details about the measuring system can be found in Brand (2005).





In a second step, samples are analyzed for $CO_2$, CO, $CH_4$, $N_2O$, $H_2$, $SF_6$ using gas chromatography. The gas chromatography (GC) system combines two Agilent 6890 gas chromatographs (Agilent Technologies Inc., USA), each equipped with combinations of sample loops, separation columns, and detection units. One of the GCs is suited for the detection of $CO_2$, $CH_4$ (applying flame ionization detection (FID)), and $N_2O$ (using electron capture detection (ECD)). The second GC is equipped with detectors for $H_2$ (pulse-discharge detector), CO (Reduction Gas Detector (RGD)), and $SF_6$ (ECD). Calibration of the measurements is based on a set of WMO tertiary reference standards and the trace gas data are reported on the respective WMO mole fraction scales. Details on the chromatographic methods and data quality assessments are provided in Heimann et al. (2022) and Worthy et al. (2023).

In the third step, samples are measured for stable isotopes in $CO_2$ and $CH_4$. High-precision analyses of stable isotopes in these gases are performed routinely at the IsoLab of the MPI-BGC (Werner et al., 2001; Ghosh et al., 2005; Wendeberg et al., 2013; Brand et al., 2016). Flasks are mounted in a fully automated cryogenic extraction system (BGC-Air Trap) that extracts $CO_2$ and $CH_4$. Isotopes of atmospheric $CO_2$ are analyzed on one of two MAT252 IRMSs, while $CH_4$ isotopes are analyzed on a system that consists of two IRMSs (Delta-V Plus, Thermo Fisher, Bremen, Germany) coupled to an extraction setup. Isotopes of $CO_2$ are reported on the JRAS-06 scale (Wendeberg et al., 2013). $\delta^{13}C-CH_4$ data is reported on the VPDB-LSVEC scale while $\delta^2H-CH_4$ isotopes are reported on the VSMOW/SLAP scales (Sperlich et al., 2016; Brand et al., 2016). Measurement uncertainties are propagated to include both individual measurement and scaling uncertainties.

In the last step of the measurement cycle, the samples are analyzed for $^{14}C$ in $CO_2$. This is done as the last step because it does not require a minimum gas pressure in the flasks, but only a minimum amount of carbon of 0.6 mg C. To analyze the $^{14}C$ concentration, the $CO_2$ is first extracted using the Air-$CO_2$-Extraction System (ACES), which consists of a manifold for 20 flasks and a cryogenic water trap (Steinhof et al., 2004). After passing the water trap, the $CO_2$ is separated from the other gases (nitrogen, oxygen, and argon) in the cryogenic trap of the Universal Gas Collection System (UGCS) (Steinhof et al., 2017). The $CO_2$ is then injected into the reaction vessel together with hydrogen. The reaction vessel is heated to 550°C so that the hydrogen reacts with the $CO_2$ to form graphite using iron as a catalyst. The graphite is then pressed into aluminum targets and loaded into magazines for accelerator mass spectrometry (AMS) measurements, which are performed with a MICADAS AMS system (Ionplus AG, Switzerland).

### 2.3.3 Data processing and release

Data on mole fractions and stable isotope ratios are ingested into the data-processing system of the atmospheric flask sampling program of the MPI-BGC (Heimann et al., 2022). This monitoring program includes 12 additional stations, in which the same gas species are measured and reported. All data from this program goes through a quality control process, in which flasks that do not meet a set of quality assurance criteria are flagged. Additional variables such as Atmospheric Potential Oxygen (APO) are computed and reported as part of a regular report released by the MPI-BGC flask program (Heimann et al., 2022).

Flags for potentially unreliable data are produced by running a test that identifies outliers based on a smoothing spline function fitted to each individual gas species. Through an iterative process, a cubic smoothing-spline function (`smooth.spline` in base R, R Core Team, 2024) is fitted first to all data points, and observations are flagged if they lie in an interval beyond



**Table 1.** Set of trace-gas species measured in the ATTO flask monitoring program. Measurement methods include gas chromatography (GC), mass spectrometry (MS), and accelerator mass spectrometry (AMS). Modified after Heimann et al. (2022).

| Gas species | Method | Analytical Precision | WMO target | Unit |
|---|---|---|---|---|
| $CO_2$ | GC | 0.07 | 0.1 | ppm |
| CO | GC | 0.8 | 2 | ppb |
| $CH_4$ | GC | 1.4 | 2 | ppb |
| $N_2O$ | GC | 0.17 | 0.1 | ppb |
| $H_2$ | GC | 0.7/2.5 | 2 | ppb |
| $SF_6$ | GC | 0.03 | 0.02 | ppt |
| $\delta^{13}C-CO_2$ | MS | 0.02 | 0.01 | ‰ |
| $\delta^{18}O-CO_2$ | MS | 0.02 | 0.05 | ‰ |
| $O_2/N_2$ | MS | 4 | 2 | permeg |
| $Ar/N_2$ | MS | 8 | – | permeg |
| $\delta^{13}C-CH_4$ | MS | 0.1 | 0.02 | ‰ |
| $\delta^2H-CH_4$ | MS | 1 | 1 | ‰ |
| $\Delta^{14}C-CO_2$ | AMS | 2 | 0.5 | ‰ |

three times the root mean square error of the residuals (Heimann et al., 2022). These observations are removed and the spline function is fitted until there are no outliers left. All observations identified as outliers are flagged with a value of 1 and reported in the main database. Observations that pass this test are flagged with a value of 0.

We use the root mean squared deviation (RMSD) between observations and the smoothing spline as an estimate of the variability of the data, accounting for seasonal variability and trends. The RMSD is obtained as the squared root of the sum of squared residuals between observations and predictions by the spline function.

    For the ATTO flask program, we combine all data on mole fractions and isotopes with data on atmospheric radiocarbon, and release an expanded report that includes all gas species measured at the site. The report includes uncertainty estimates for each

measurement obtained from each laboratory's procedure for quantification of uncertainty based on each gas' set of standards and blanks. The final report is released on the ATTO Data Portal (https://www.attodata.org), and each data release is assigned a digital object identifier (doi). We aim at producing at least one data release per year, assuming no major delays in the flask cycle.

## 3   Results

This article reports the release of version 2025.1 of the dataset (Sierra et al., 2025), which contains data between the sampling dates of 2021-09-09 and 2024-01-04. This data release contains a total of 117 flasks have gone through the complete cycle of



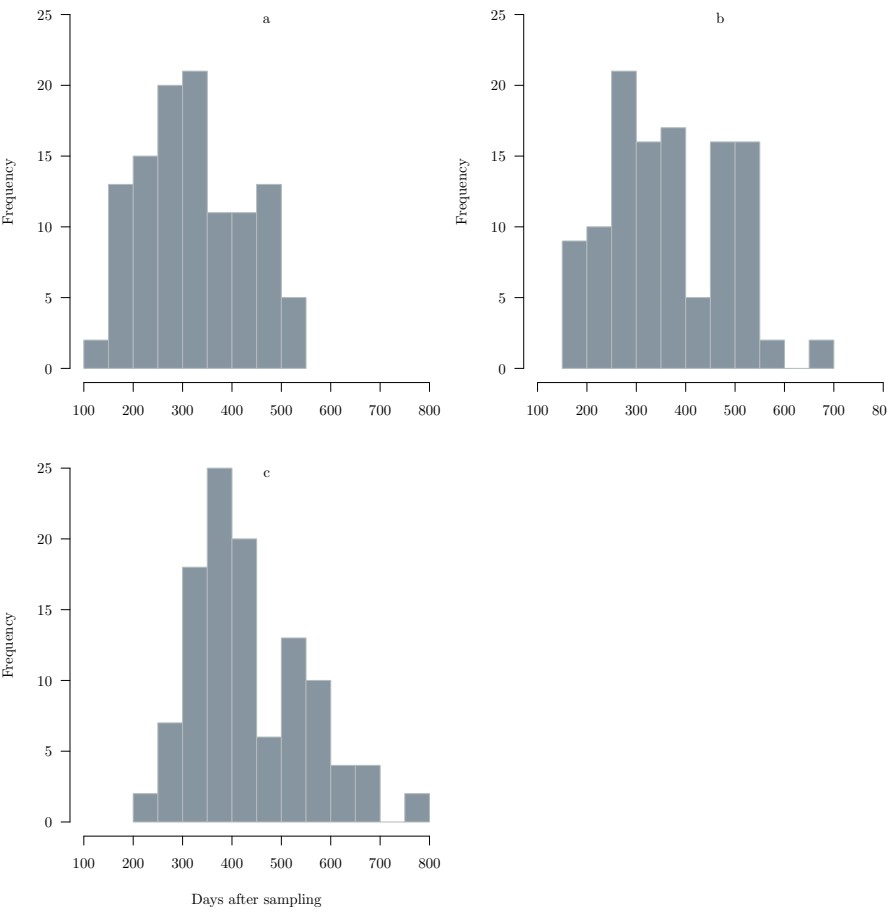

**Figure 3.** Frequency distributions of storage times of flasks in days since sampling until laboratory analysis for (a) $O_2/N_2$ and $Ar/N_2$ ratios, (b) GC measurements of trace gases, (c) IRMS measurements of stable isotopes.

sample collection, analysis and data reporting. In this article, we report only observations taken at mid-day, but all observations, including those taken at night, are also provided in the released dataset.

Due to the remoteness of the field site, and the complexity of the logistics for sample transport between Brazil and Germany, the samples remain stored in the flasks for a minimum of three months and up to 26 months before analyses (Figure 3). On average, the storage time until first measurements of $O_2/N_2$ and $Ar/N_2$ ratios is 324 days. For measurements of mole fractions by GC, the average storage time is 365 days; and for stable isotope measurements by IRMS the average storage time is 435 days. The last measurements of $^{14}C-CO_2$ by AMS take even longer, with average storage times of about 500 days.



## 3.1 Gas chromatography measurements

Initial results obtained from measurements by GC in the flasks samples are shown in Figure 4. The time series for $CO_2$ mole fraction showed no clear seasonal cycle and a slight upward trend. Only one observation from this series was flagged as anomalous, and the root mean squared deviation (RMSD) of the observations with respect to the smoothing spline gave an estimate of variability of the observations of 3.9 ppm.

The time series for $CH_4$, $N_2O$, and $SF_6$ mole fractions also showed an upward trend and some degree of seasonality, with
lower mole fractions during the dry season and higher values in the rainy season. The RMSD for $CH_4$ was 19.8 ppb with no detected outliers. For $N_2O$ and $SF_6$, the variability around the smoothing spline was much smaller, 0.21 ppt and 0.04 ppt, respectively. One anomalous observation from the same flask was detected for $N_2O$ and $SF_6$ (Figure 4).

For CO, strong peaks were observed in the dry seasons of the years 2022 and 2023, likely related to local fires that are common during the dry and hot season (Andreae et al., 2012). No anomalous observations were detected for CO and the
RMSD with respect to the smoothing spline was 13.3 ppb.

Observations of $H_2$ showed relatively high variability in comparison to other gases, with a smoothing spline curve with double peaks within one year and six anomalous observations (Figure 4). The RMSD for $H_2$ with respect to the smoothing spline was 9.05 ppb.

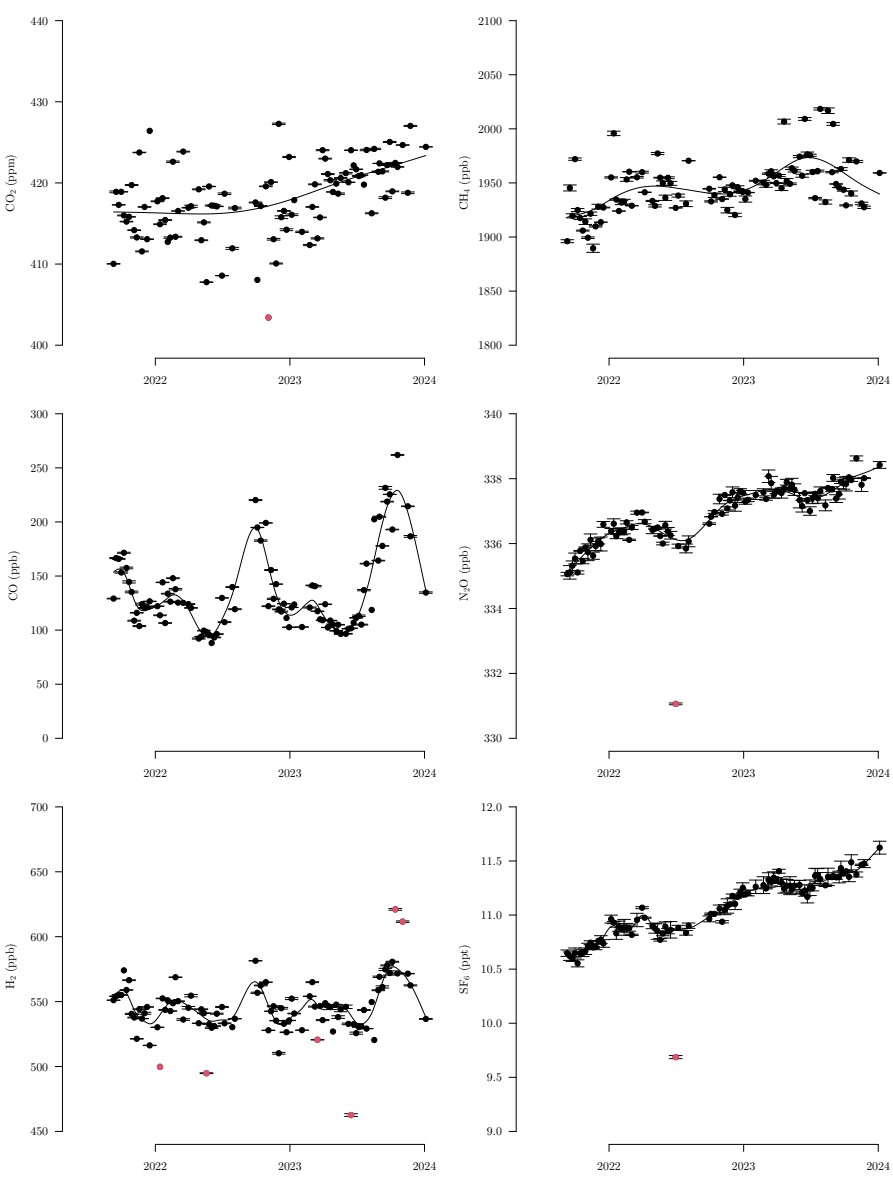

**Figure 4.** Gas species measured by gas chromatography in flask samples from the Amazon Tall Tower Observatory. Lines represent smoothing cubic-splines fitted to all data points of individual species through an iterative process that removes outliers until all observations left are inside the range of three times the root mean squared deviation. Outliers, represented as red points, are not reliable observations for determining background conditions at the local level. Error bars represent analytical uncertainty.





## 3.2 Mass spectrometry measurements

Values of $\delta^{13}C-CO_2$ showed a small declining trend over time, larger in 2023 than in 2022, with no clear seasonal cycle (Figure 5). Two observations with enriched $^{13}C$ were flagged as outliers and the RMSD with respect to the smoothing spline was 0.17 ‰. The values of $\delta^{18}O-CO_2$ showed a marked level of seasonality, but no clear upward or downward trend. Enriched values of $^{18}O$ were observed at the end of the dry season, and depleted values in the rainy season. Two observations of $\delta^{18}O-CO_2$ were flagged as outliers and the RMSD was 0.49 ‰.

The stable isotopes of methane showed a weak declining trend over time (Figure 5), and no strong seasonal cycles. Four anomalous observations were flagged for $\delta^{13}C-CH_4$ while no outlier was flagged for $\delta^2H-CH_4$. The RMSD for $\delta^{13}C-CH_4$ was 0.22 ‰, and for $\delta^2H-CH_4$ 2.48 ‰. Similarly for the $Ar/N_2$ ratios, which showed no clear trend over time and a weak seasonal cycle masked by relatively large variability around the smoothing spline (RMSD 35.3 permeg).

The values of $O_2/N_2$ showed a more consistent declining trend, accentuated in the year 2023 (Figure 5). Only one observation 215 was flagged as outlier and the RMSD was 20.9 permeg.

## 3.3 Accelerator mass spectrometry

Values of $\Delta^{14}C-CO_2$ also showed a significant level of variability with no clear seasonality or decreasing trend (Figure 6). The values of $\Delta^{14}C-CO_2$ were relatively high in the dry season of 2021 and varied at a value close to 0 ‰ after 2022. Two observations in 2023 showed a significant depletion in radiocarbon with respect to the entire series. Although global 220 atmospheric radiocarbon reached a value close to 0 ‰ around the years 2020-2021 (Graven et al., 2022), more or less consistent with our observations, the dilution of $\Delta^{14}C-CO_2$ by the combustion of fossil fuels is expected to consistently dilute these values over time (Graven, 2015). We do not see this continuous dilution effect in our data, but rather a convergence of $\Delta^{14}C-CO_2$ to a stationary value close to 0 ‰.

## 3.4 Surface influence footprints

As reference for the interpretation of the data, we provide here mean surface influence trajectories obtained from the Stochastic Time-Inverted Lagrangian Transport model STILT (Lin et al., 2003) with a similar setup as in Botía et al. (2022), but here for an arrival altitude of 324 m agl. Trajectories backward in time, computed as monthly averages between September 2021 to February 2023, show that the footprint of the gases arriving at the ATTO tower are mostly from easterly directions (Figure 7). During the dry-season months, from June to October, the footprint is mostly from the easterly to southeasterly directions. In 230 the rainy season months, from November to March, the footprint is mostly from easterly to northeasterly directions.

These footprints are consistent with previous analyses presented by Pöhlker et al. (2019) and Botía et al. (2022), as well as the measurements of $SF_6$ (Figure 4) that show a decrease in mole fraction during the dry season and an increase in the rainy season. Because $SF_6$ is mostly produced in the northern hemisphere, larger values indicate contributions from northern hemisphere air and lower values a larger contribution from southern hemisphere air. This implies that the flask measurements at

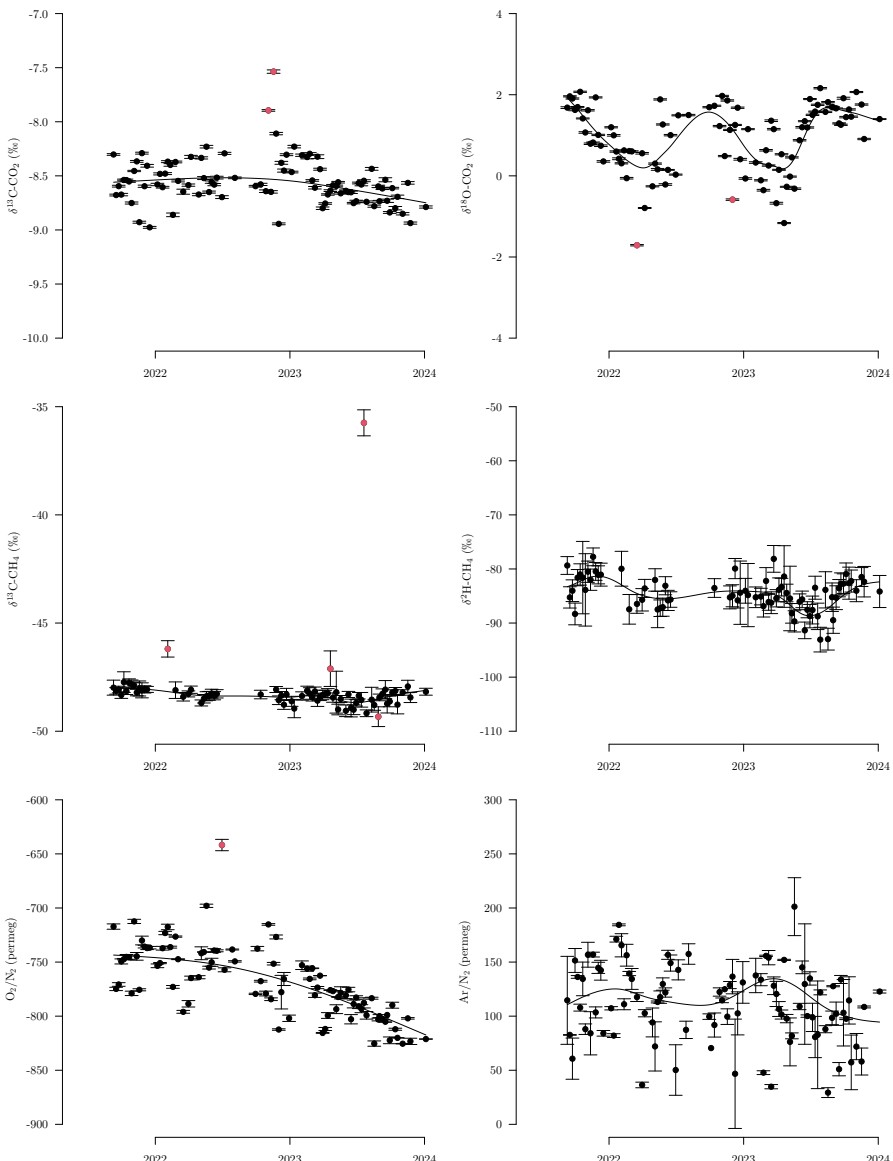

**Figure 5.** Gas species measured by mass spectrometry in flask samples from the Amazon Tall Tower Observatory. Lines represent smoothing cubic splines fitted to all data points of individual species through an iterative process that removes outliers until all observations left are inside the range of three times the root mean square error. Outliers, represented as red points, are not reliable observations for determining background conditions at the local level. Error bars represent analytical uncertainty.

ATTO have a stronger northern hemisphere influence during the rainy season, and that the dry season is more strongly affected by airmasses coming from the southern hemisphere, in agreement with the analysis of Andreae et al. (2012).



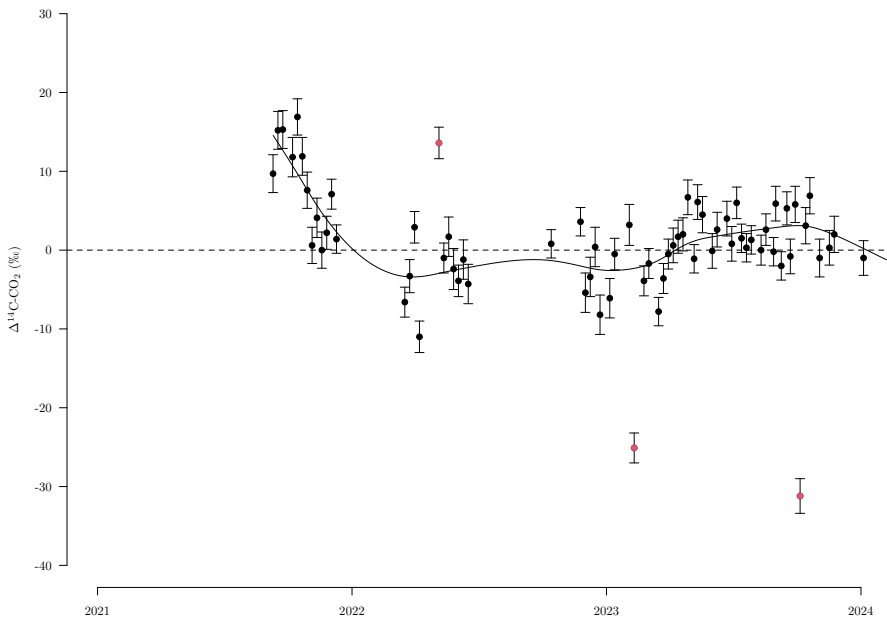

**Figure 6.** Radiocarbon in atmospheric carbon dioxide ($\Delta^{14}C-CO_2$) measured in flasks at a height of 324 m agl at the Amazon Tall Tower Observatory. Error bars represent analytical uncertainty.

## 4 Discussion

The new flask monitoring program established at the Amazon Tall Tower Observatory provides a comprehensive assessment of trace gases, stable isotopes, and radiocarbon in atmospheric $CO_2$. This system is unique in South America and in the Amazon

forest region, and it provides important information to support other studies that require information on background conditions for a number of gas species at the local level. Previous gas monitoring systems based on aircraft measurements have provided invaluable measurements of main GHGs such as $CO_2$, $CH_4$ and CO (Gatti et al., 2014, 2021). The new monitoring system at ATTO expands the number of gas species and isotopes being measured in this region considerably, and offers an opportunity for integrated assessments using both monitoring systems. Furthermore, other continuous in-situ measurements at ATTO (e.g.

Botía et al., 2022) also provide new opportunities for comprehensive assessments of land-atmosphere interactions both at the site and at the regional levels.

Given the complexity of logistics for the transport and analysis of samples, the storage time of samples is currently higher than one year for many samples. This may potentially have some influence on the quality of the measurements (Sturm et al., 2004), particularly for CO, but effects of storage time are minimized in our case by the use of PCTFE sealing in flasks (Rothe

et al., 2005). The large emission pulses of CO that have been observed during the dry seasons (Figure 4), may diminish the negative effects of storage time on this gas given the large fluctuation signals observed relative to potential biases. The large majority of the measurements we have obtained until now pass all quality tests established by the MPI-BGC flask monitoring



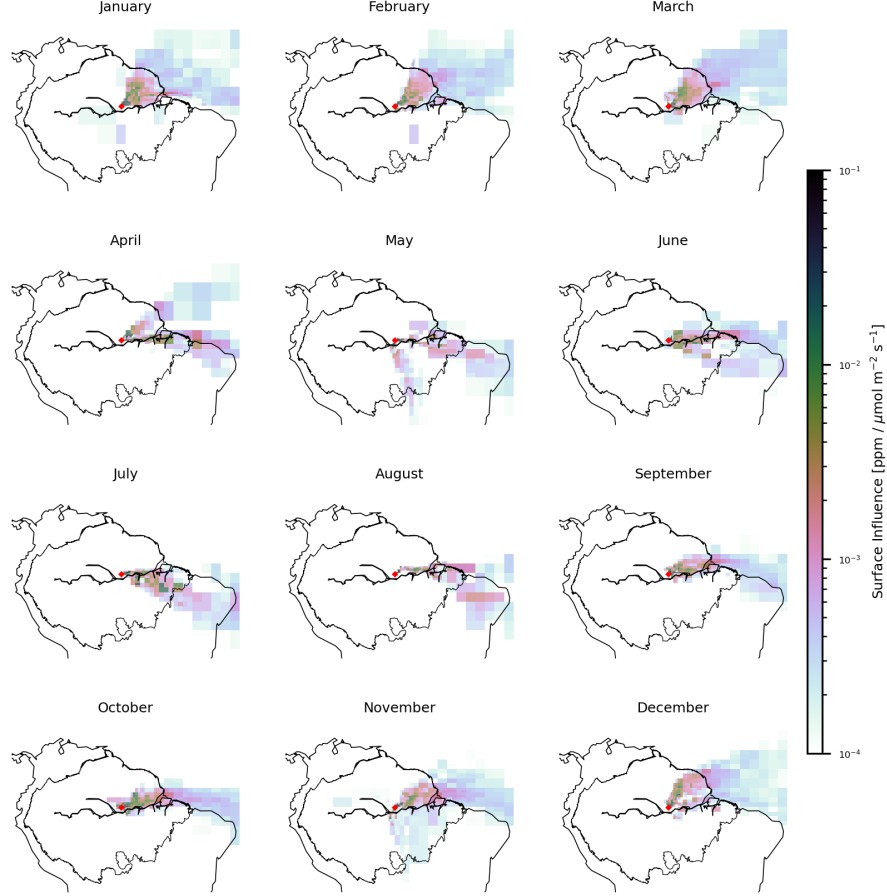

**Figure 7.** Mean surface influence for each month and then averaged over years (2021-Sep to 2023-Feb). The surface influence was generated using the STILT model at the location of the Tall Tower at the ATTO site (-2.14, -58.99). The black contour indicates the biogeographic Amazon limits.

program. As our monitoring program at ATTO advances over time, we anticipate that logistical challenges will diminish, leading to shorter sample storage times. This applies in particular to the processing of import/export permits for samples, which in some cases have taken considerable time. This aspect has recently improved, and with a more routine procedure for obtaining transport permits, we expect the average storage time to decrease by approximately three months in the future.

For some gas species such as $CO_2$, $CH_4$, and their stable isotopes, we did not observe a strong seasonal cycle. Although we did observe an increasing trend for the mole fraction of these gases, the variability observed was relatively large, particularly for $CO_2$. As opposed to other monitoring sites in relatively clean atmospheres such as Mauna Loa, South Pole, or Alert (Keeling and Graven, 2021; Heimann et al., 2022), the footprint at ATTO is strongly influenced by large biogenic sources from the Amazon forest on the east side of the tower (Figure 7) (Pöhlker et al., 2019; Holanda et al., 2020; Botía et al., 2022). As



opposed to high latitude regions, where seasonal growth of vegetation is very marked, the evergreen tropical forest vegetation has a much weaker seasonal growth cycle, only modulated by seasonal cycles in precipitation and fire activity. We observed a strong seasonal cycle in CO related to fire occurrences in the dry season. We also observed small seasonal cycle in $N_2O$, with lower values in the dry season and higher values in the rainy season.

The observed seasonal cycle of $SF_6$ is synchronous with the occurrence of the dry and rainy seasons. As $SF_6$ has no biogenic sink or source, and it is mostly emitted by anthropogenic activities in the northern hemisphere, its seasonal cycle provides good support to the idea that dry and rainy seasons in the study region are mostly a consequence of large scale atmospheric dynamics, particularly the movement of the inter-tropical convergence zone (ITCZ) (Pöhlker et al., 2019; Andreae et al., 2012).

Radiocarbon in atmospheric $CO_2$ does not show a clear trend or seasonal cycle. Although we did not expect to observe a strong seasonal cycle during the dry and rainy season because plant and microbial respiration are active most of the year, we did expect a more clear decrease of $\Delta^{14}C-CO_2$ over time. Fossil fuel emissions are decreasing $\Delta^{14}C-CO_2$ globally, and we expected a similar trend here. However, it seems that the combined effect of ecosystem respiration and fire emissions, which are enriched in radiocarbon (Chanca et al., 2025), is counteracting the fossil-fuel dilution trend.

The time series presented in this article are still relatively short, and it is difficult at this point to reach strong conclusions on trends, sources, and sinks of the gas species measured. However, this is an ongoing monitoring program and the time series will grow over time. All data produced in this monitoring program is being released openly after appropriate data curation and processing.

## 5 Conclusions

A new long-term monitoring program of greenhouse gases and isotopes has been established at the Amazon Tall Tower Observatory with a footprint that covers a significant part of the central and eastern part of the Amazon River basin. This program currently produces data on the mole fractions of major greenhouse gases ($CO_2$, $CH_4$, $N_2O$, CO, $SF_6$), stable isotopes of $CO_2$ and $CH_4$, radiocarbon in $CO_2$, and other gases of interest, such as $H_2$, as well as $O_2/N_2$ and $Ar/N_2$ ratios. The program started in September 2021, and until now data from 117 flasks have been released under an open access license.

The current set of measurements shows an increasing trend for many gas species such as $CO_2$, $CH_4$, $N_2O$, and $SF_6$. In general, there are no marked seasonal cycles for most gas species, as it is frequently observed at northern hemisphere stations. However, some observed seasonality is related to dry and rainy seasons. The seasonal dynamics of fire and the movement of the ITCZ also play an important role in the observed variability for many gas species.

We expect that the data produced by this program would be of interest for many other studies that rely on precise background data, particularly for isotopic studies. The ATTO project is committed to continue these measurements over time and make them openly available to the research community and the general public.





# 6 Code and data availability

The data set described in this article is available in the ATTO data portal, https://doi.org/10.17871/ATTO.465.13.1902 (Sierra et al., 2025). Code to reproduce data figures presented in this article is available at https://gitlab.gwdg.de/carlos.sierra/atto-fmp.
295 git. Upon manuscript acceptance, this repository will be move to a permanent archive in Zenodo with a respective doi.

*Author contributions.* CRediT Taxonomy of author contributions as follows. Conceptualization: CAS, MA, ACA, LAC, ME, LVG, SH, MH, AJ, JVL, IL, CAQ, ST, SZ; Data curation: CAS, IC, HvA, SB, CQDJ, ME, LVG, MG, SH, AJ, IL, JM, HM, MR, CR, AS; Formal analysis: CAS, IC, HvA, SB; Funding acquisition: CAS, MA, ACA, LAC, LVG, MH, JVL, IL, KM, CAQ, BT, ST; Investigation: CAS, IC, LB, SB, CSCC, ME, AF, LVG, SH, AJ, SK, RK, YS; Methodology: CAS, IC, LB, SB, CSCC, ME, AF, LVG, SH, MH, AJ, SK, RK, JVL, IL, HM,
300 MR, CR, YS, AS; Project administration: VH, JVL, CAQ, BT, ST; Software: CAS, ME, MG, MH, JM, CR; Resources: ME, LVG, MH, AJ, RK, IL, KM, HM, CAQ, MR, AS, BT, ST, SZ; Supervision: CAS, LVG, MH, AJ, IL, KM, HM, CAQ, AS, BT, ST, SZ; Validation: IC, HvA, LB, SB, CSCC, CQDJ, AF, LVG, SH, AJ, SK, IL, HM, MR, CR, AS; Visualization: CAS, HvA, ME, CR; Writing – original draft: CAS; Writing – review & editing: CAS, IC, MA, HvA, SB, CSCC, CQDJ, HM, AS, SZ;

*Competing interests.* The authors declare no competing interest.

305 *Acknowledgements.* The Amazon Tall Tower Observatory (ATTO) is a research infrastructure funded by the governments of Brazil and Germany. Funding for the collection and processing of the data presented in this article was mostly provided by the German Federal Ministry of Education and Research (grant numbers 01 LK 1602 C and 01 LK 2101 A) and the Max Planck Society. We thank C. König and R. de Souza for valuable logistic support. Special thanks go to the personnel that maintains operations at the ATTO research station, which includes: A. H. Melo Nascimento, A. Rodrigues Pereira, N. A. de Castro Souza, S. Bulthuis, and V. Ferreira de Lima.



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
