# Peer review of "The flask monitoring program for high-precision atmospheric measurements of greenhouse gases, stable isotopes, and radiocarbon in the central Amazon region"

_Earth System Science Data, 2025_

## Referee Comment (RC1)

**Review of manuscript submitted to ESSD**

The flask monitoring program for high-precision atmospheric measurements of greenhouse gases, stable isotopes, and radiocarbon in the central Amazon region (Sierra et al.)

This paper presents an update to the flask sampling dataset from the ATTO tower. This is an excellent contribution of data from a data poor region and it is admirable that the authors are making the data available quickly for the great benefit of the scientific community. The metadata is very well represented in the data download. The description of the methods is clear and the brief analysis of the main trends is interesting. I can recommend publication once the following points are addressed, in particular regarding the quality of the figures.

**Comments:**

- Figure 2 could include additional information. I understand you want to keep it relatively simple but all the same to serve as a useful overview it would be good to add at least:
  - o Brief details of the flask sampling (frequency, time, volume, height)
  - o Clear link to where the data can be found (data repository)
  - o Info about how often "finalized" data is released
  - Brief info about QA/QC procedure
- Also, the formatting is a bit odd, such as the hyphen in 13C-CO2 which is much too long, 14C with no superscript, and other "boxes" which could be much wider for a nice figure. The colours are a bit dark also for MPI-BGC and Jena, making it hard to read the text.
- Section 2.3.2: Are the analyses on flasks always done in this order? Is there indication whether there is any isotopic fractionation in the first two sets of measurements, that could affect the isotopic composition measured in the third step? Has this been investigated?
- Section 2.3.3: What are the quality assurance criteria? How often are these not met?
  - Regarding the flagging based on deviation from spline: Are there any species that are highly variable at this timescale, so that flagged observations could be true variability? Eg. for H2 you have much more flags.
- **Figure 3:** Add more info on the figure panels so that it can be more easily understood, eg. the info that the panels represent air components, trace gases, and IRMS measurements respectively could be on the figure. The text is very small on the axes. The placement of the subplot designator (a,b,c) is very unusual; please move to the top left and perhaps make it bold so it stands out. I think the x axis would be more useful if it was in months, not days.
  - Also regarding this section: Do you have some context regarding whether there are any potential issues with the long storage times?
- **Figure 4**: The axis and tick labels are way too small. The red colour of the outliers is hard to distinguish because the points are so small and the error bars are black not red. Please make the x axis span the full length of the subpanels.
  - Could you add the trendline (where there is a trend) as well as the spline?
  - Same comments for figures 5 and 6.
  - Data: A few small points could be altered to improve usability of the data:
    - Remove spaces in column titles, eg. SD CO2 could be SD\_CO2

- Use a standard designation for error, eg. you have SD CO2 and err\_O2/N2. If both are measurement uncertainty they should be designated the same way.
- o Remove protected characters like /
- Have the time in a less error-prone format, with no / eg. just "18112021 16:30" or similar

**Minor points:**

- L9: Represent stable isotopes as delta values, eg. change  $^{13}\text{C-CO}_2$  to  $\delta^{13}\text{C-CO}_2$  and analogously for other species.
- L208: Of  $\delta$ 180 not of 180.

---

## Author Response (AR1)

**Max-Planck-Institut für Biogeochemie**

Max Planck Institute for Biogeochemistry

MPI für Biogeochemie · Postfach 10 01 64 · 07745 Jena, Germany

Tobias Gerken Editor Earth System Science Data

Dr. Carlos A. Sierra Tel.: +49-(0)3641-57-6133 csierra@bgc-jena.mpg.de

23rd July 2025

Dear Editor,

Thanks for your support with this article.

We have provided answers to all comments from the reviewers, and prepared a new version of the article with improved figures and more clear description of the quality assurance and flagging procedures. We also prepared a new version of the dataset addressing reviewers' comments and adding recent data.

You can find point-by-point answers to all reviewers' comments below. These are the same answers that were already provided in the discussion forum.

**Reviewer 1**

This paper presents an update to the flask sampling dataset from the ATTO tower. This is an excellent contribution of data from a data poor region and it is admirable that the authors are making the data available quickly for the great benefit of the scientific community. The metadata is very well represented in the data download. The description of the methods is clear and the brief analysis of the main trends is interesting. I can recommend publication once the following points are addressed, in particular regarding the quality of the figures.

We thank the reviewer for appreciating the value of our work and for the helpful comments to improve our article and the dataset.

Figure 2 could include additional information. I understand you want to keep it relatively simple but all the same – to serve as a useful overview it would be good to add at least: - Brief details of the flask sampling (frequency, time, volume, height) - Clear link to where the data can be found (data repository) - Info about how often "finalized" data is released - Brief info about QA/QC procedure

We added the sampling details and the link to the data repository to the figure as suggested. We did not add information on how often finalized data is released because this occurs at very irregular intervals that are difficult for us to anticipate

given the complex logistics of this program. The QA/QC information is lengthy, and described in its own paragraph in section 2.3.3.

Also, the formatting is a bit odd, such as the hyphen in 13C-CO2 which is much too long, 14C with no superscript, and other "boxes" which could be much wider for a nice figure. The colours are a bit dark also for MPI-BGC and Jena, making it hard to read the text.

We fixed these different formatting issues in the figure and added other improvements.

Section 2.3.2: Are the analyses on flasks always done in this order? Is there indication whether there is any isotopic fractionation in the first two sets of measurements, that could affect the isotopic composition measured in the third step? Has this been investigated?

The flasks are always analyzed in the same order. Crucially the O2/N2 and Ar/N2 measurements need to be done first. Due to the larger mass difference of the analyzed molecules in the ratio, fractionation can occur here due to micro-leaks or improper flask handling, and indeed the measurement itself. The relative mass difference between the molecules that are analyzed during the classical isotope ratio analyses, e.g. 13CO2 and 12CO2 is much lower, so that relative diffusion speeds of the species play a much smaller role. This has been tested by filling flasks with air and doing the same measurement on the same flask repeatedly to see the effect of flask handling, sample pressure reduction and day to day variability. These tests have shown that the largest contributor to measurement uncertainty is the sample pressure. The analyses of d13C and d2H isotope signatures of methane reveal good repeatability at sample pressures above 1200 mbar. At lower pressures the repeatability deteriorates. This threshold is 1400 mbar for O2/N2 measurements. Flasks that do not have the required sample pressure are either flagged or not analyzed at all.

Section 2.3.3: What are the quality assurance criteria? How often are these not met? Regarding the flagging based on deviation from spline: Are there any species that are highly variable at this timescale, so that flagged observations could be true variability? Eg. for H2 you have much more flags.

Thanks to this comment, and to other comment from Reviewer 2, we realized that we had a problem with our reporting of the QA/QC and flagging procedure. We have two separate flagging systems, one due to QA by the measurements in the laboratory, and another by running a statistical test to find points that are above 3 times the total uncertainty of the time series. In the previous version of the manuscript, we did not consider the first flagging, which resulted in one measurement included in the released database that should have not been included due to flagging from the lab's QA. We prepared a new release of the data and removed this measurement. The flags in the new released dataset are only due to the statistical check, and therefore the color coding in the figure has changed.

The flagging presented in the released dataset and in the figures is thus only due to natural variability. This flagging convention is consistent with the system used for releasing the flask data from the larger flask monitoring program of the Max Planck Institute for Biogeochemistry (https://doi.org/10.17617/3.V2EXHD). To

clarify this issue, we added text in section 2.3.3. to more explicitly tell readers that these flags are only due to statistical variability.

A short description of the internal QA flagging system was included in section 2.3.3.

Figure 3: Add more info on the figure panels so that it can be more easily understood, eg. the info that the panels represent air components, trace gases, and IRMS measurements respectively could be on the figure. The text is very small on the axes. The placement of the subplot designator (a,b,c) is very unusual; please move to the top left and perhaps make it bold so it stands out. I think the x axis would be more useful if it was in months, not days.

This figure has been modified following reviewers' suggestions. We changed the time unit to months, moved the labels to top left and made them bold, and added the gas species names corresponding to each panel.

- Also regarding this section: Do you have some context regarding whether there are any potential issues with the long storage times?

Effects of storage times were previously investigated by Rothe et al. (2005), who found that the most critical aspect influencing the measurements is the type of sealing material in the flasks. Over time, trace gases may diffuse over the O-rings that seal the flasks. Tests conducted at our laboratories, found that fluoropolymer (PFA) material is not well suited for the O-rings because trace gases tend to diffuse over time, particularly for CO. O-rings made of polychlorotrifluoroethylene (PCTFE) gave better results in these previous tests and for this reason we use PCTFE in the sealing of our flasks. To further tests storage time effects, we are currently comparing our measurements with the data simultaneously measured at the same sampling height by a FTIR instrument. The results from those comparisons would be published in detail in a separate article.

Figure 4: The axis and tick labels are way too small. The red colour of the outliers is hard to distinguish because the points are so small and the error bars are black not red. Please make the x axis span the full length of the subpanels. - Could you add the trendline (where there is a trend) as well as the spline? - Same comments for figures 5 and 6.

All figures were redrawn based on these comments. Labels are larger, error bars redesigned, color palette improved for contrast.

Data: A few small points could be altered to improve usability of the data: - Remove spaces in column titles, eg. SD CO2 could be SD\_CO2 - Use a standard designation for error, eg. you have SD CO2 and err\_O2/N2. If both are measurement uncertainty they should be designated the same way. - Remove protected characters like / - Have the time in a less error-prone format, with no / eg. just "18112021 16:30" or similar

We fixed all these issues identified by the reviewer and released a new version of the data. For this reason, we provide now a new link to dataset due to the change in its structure. This new release also includes a few more data points that became available during the last weeks.

Minor points: - L9: Represent stable isotopes as delta values, eg. change 13C-CO2

to d13C-CO2 and analogously for other species.

Done.

- L208: Of d18O not of 18O. Done.

References: Rothe, M. T., Jordan, A., and Brand, W. A.: Trace gases,  $\delta 13\mathrm{C}$  and  $\delta 18\mathrm{O}$  of CO2-in-air samples: Storage in glass flasks using PCTFE seals and other effects, in: Proceedings of the 12th IAEA/WMO meeting of CO2 experts, Toronto, Sept. 2003, pp. 64–70, https://hdl.handle.net/11858/00-001M-0000-000E-D36E-0, 2005.

**Reviewer 2**

The paper describes a very impressive and unique data set. The data is obviously of excellent quality, and a lot of care and work has gone into it. Being very familiar with the challenges of flask sampling and the logistics involved, I know that this is a truly unique effort. I specifically admire that the measurements are made on a continuous and long-term basis. The paper itself is written in simple, clear English that I found easy to read and understand. I would have only a few minor stylistic comments and suggestions if it were not for the data filter used to determine "unreliable data points." I chose major revisions, simply because I think this point needs to be reviewed again.

Thanks for recognizing the value of the data and the effort we put into it. We realized that there was confusion with our reporting of the flagged data, and made corrections accordingly to clarify this issue.

From the last paragraph, I get the impression that maybe the authors are more used to so-called background sites. The filtering method used might potentially be appropriate at a site where minimal variability is expected and where high-frequency observations are available. Even then, I would probably rename the flag to a pollution flag or non-baseline flag. At a site with more complex atmospheric influence such as this one, statistically filtering the data may remove realistic and normal variability. In weekly flask data, there is not necessarily a lot of connection between each individual observation point; sources/sinks and meteorological conditions may change at a higher frequency than that.

We agree with these comments, but we would also like to clarify that these measurements are part of a larger network of flask measurements conducted at a number of so-called background sites. Our QA/QC and flagging system is consistent across the network, and instead of changing the flagging criteria only for this site, we included some clarifications in the text to better interpret the flagged data.

In particular, the data processing system for the entire flask network has two flagging components, one internal QA test where problematic measurements are flagged, and a second test based only on the statistical variability of the data. Measurements flagged internally are removed from the final data release. The statistical flags are reported and released with the final dataset. In the previous version of the article, we mistakenly included an observation that was flagged

internally as not meeting the QA standards of the laboratory. This data point was not included in the final release, and it was removed from the updated release accompanying the revised version of this article.

The statistical flags are indeed plausible measurements, and they are simply flagged because they are outside the range of the observed variability in the current time series. These extreme observations are indeed plausible, and in cases such as CO, they represent spikes due to regional fire events.

In Figure 4, I find the variability of  $CO_2$  (only one flagged point) and  $H_2$  (lots of flagged points) to be fully within normal atmospheric values. I assume the statistical filter is used to determine spurious values that cannot be explained, such as the  $SF_6$  value and the  $N_2O$  value that are clearly below southern hemispheric background values (I compare my low data points to a southern hemisphere background site such as Cape Grim to determine what is realistic and what is not). I agree that those points should be flagged, as they are clearly not realistic. Those low values are probably caused by a problem during the sampling and some of the fill gas from the ICOS flask lab remaining in the flask, diluting the sample. I believe NOAA specifically uses a fill gas that is unlike atmospheric values to diagnose incomplete filling through these unrealistically low values.

The anomalous SF6 and N2O values were indeed from the same flask that was internally flagged and that was mistakenly included in the previous release. This measurement was flagged internally as not valid due to failure at retrieving the sample. The other outliers in Fig 4 were marked according to the statistical test, and therefore the previous version of this figured mixed up the two flagging systems. We apologize for this problem, and we hope that the new version of the data, figures, and text clarifies this issue.

In my opinion, this point should then be flagged in all compounds, as an incomplete filling affects all data, not just the obviously strange-looking one. For the 13C-CH4, Figure 5 shows a very enriched value around -37 permil, that I think may be unrealistic, I tried to look up its matching CH4 data point but I was unable to find that -37 permil point in the repository, I attached the plot I get. The 14C-CO2 seems realistic to me. As the authors correctly identified, there are some depleted points consistent with fossil fuel burning and some enriched values that are probably caused by biomass burning. Enriched values can also be from nuclear industry emissions, but I presume biomass burning is more likely here.

Correct. See answer to previous comment.

I do not understand the atmospheric variability of the  $O_2/N_2$ ,  $Ar/N_2$ , 18 O-CO2, and  $\delta^2 H$ -CH4 values well enough to comment on those.

I believe strongly that only data points with clear instrumental problems or points that are evidently unrealistic should be flagged. Modelers using the data set should be aware of the variability of the atmosphere and use their own judgment on how much weight an individual sampling point should be given. Just flagging that one flask with the low  $N_2O$  and  $SF_6$  value and re plotting all the data without filtering would satisfy me as a revision.

As mentioned previously, our flagging system is the same for all stations of the flask monitoring program of the Max Planck Institute for Biogeochemistry. For

this reason, we use a consistent statistical flagging system, but we warn users that this is a statistical test, and measurements are indeed plausible given the high levels of natural variability at this site. We included more details in the text to clarify this issue.

The hard work and passion that went in to this data set is evident and besides the filtering I have only very minor comments.

Thanks. Indeed, without passion we would not have done so much work!

- Line 28: Mentions these aircraft campaigns, but it is not evident in the text that the previously referenced papers are aircraft campaigns.

The word 'these' in this sentence was confusing. The correct references are the ones at the end of the sentence. By removing the word 'these', we eliminated this ambiguity.

- Line 52: "Infrastructure to access relevant heights" — Could you please reword this to make clear what you mean? At the moment, the ICOS sampler description and the rarity of having remote infrastructure with tall towers that provide appropriate sampling heights are a bit convoluted.

These sentences were reworded for clarity.

- Line 96: "Each flask has one valve at each end" — maybe: Each flask is equipped with a valve at both ends.

**Changed as suggested.**

- Figure 2: There is a car icon missing between the Isolab and the 14C lab. I think the text description in Chapter 2.3.1 is important to show how remote the site is, but it could maybe be summarized a bit more, especially for the transport back to Jena. At the moment, we have the chapter description, the graph, and the description of the graph. Maybe the description of the graph could be more along the lines of a title such as: Flow chart of shipping logistics, or the return shipment could be summarized as being the same process in reverse order?

The 14C lab is located within our own institute in the same building. There is no need to transport the samples to an external 14C lab, so the car icon is not missing from the figure.

- There is a bit of repetition in the description of the assignment of the unique sample number and the analysis in Jena as well — Line 120 and 124.

**Repetition removed.**

- Line 139-140: Could you check the references? They do not seem to contain a detailed description of the measurement setup, or alternatively, give more details about the setup in the description.

The reference Heimann et al (2022) contains the most updated description of the measurement setup.

\*- Line 149: "Measurement uncertainties are propagated to include both individual measurement and scaling uncertainties" — I think scaling probably refers to the calibration scale? So maybe something along the lines of:

- The uncertainties reported contain the individual measurement uncertainty as well as the propagated calibration scale uncertainty.
- Or: Error propagation was used to account for the measurement uncertainty as well as the propagated calibration scale error.\*

We changed the sentence according to the first option proposed by the reviewer.

- Line 162-163: Could you describe or reference the quality control process used?

We expanded this section to provide more details and to clarify the two flagging systems for the data.

\*- Line 173–174: "For the ATTO flask program, we combine all data on mole fractions and isotopes with data on atmospheric radiocarbon, and release an expanded report that includes all gas species measured at the site." — Please rephrase. Maybe something along the lines of:

All measurement results from the different instruments and labs are combined into one report.\*

Changed as suggested.

- Line 235–236: The formatting is a bit awkward, as there are just two text lines on the page with Figure 5.

These two text lines were moved to a separate page.

We hope this new version addresses well all reviewers'c comments and is now suitable for publication in ESSD.

Best regards,

Carlos A. Sierra, on behalf of all coauthors

---

## Author Response (AR2)

**Max-Planck-Institut für Biogeochemie**

Max Planck Institute for Biogeochemistry

MPI für Biogeochemie · Postfach 10 01 64 · 07745 Jena, Germany

Tobias Gerken Editor Earth System Science Data

Dr. Carlos A. Sierra Tel.: +49-(0)3641-57-6133 csierra@bgc-jena.mpg.de

5th September 2025

Dear Editor,

Thanks for your support with this article. We created a new version of the dataset following the recommendations from the second round of review. The new submitted manuscript contains only minor changes to the text describing the identification of outliers and removing text on the previous report of data flags. In the following, you can find a detailed response to the comments from reviewer 2.

**Reviewer 2**

Unfortunately, the revisions do not sufficiently address the issue of the smoothing function flag. I will try to explain the problem in more detail. The MPI-BGC network referred to in the response does contain a flag column; the flag column description in the header of the file reads: good flask replicate average, possibly not representative for large-scale background. While I fundamentally disagree with adding interpretation to observational datasets, this description is very clear and easy to understand. In this work, the values are still described as outliers, and a user would have to very carefully read the entire paper to understand that all data has passed quality assurance and reflects normal variability. In the referenced database that contains the dataset, the flag description is not immediately apparent (it is a click away), and it is misleading. The description still refers to the data that is flagged as anomalous. Most users would assume they should only use the unflagged part of the dataset. All methods to determine baselines from observations are a matter of choice; there is no correct or standard approach. Forcing this choice on the data user, even unintentionally, could lead to biases in their work. In my opinion, the simplest way to avoid confusing what is a baseline flag column with a bad data quality flag is to remove the column entirely. At the very least, the nomenclature and description of that baseline flag would have to be much more neutral (e.g., possibly not background, as in the MPI-BGC network referenced). The description would have to be consistent within the dataset and the paper. Even if a user only downloads the dataset, the fact that this is an interpretation and not a data quality flag needs to be immediately apparent. I am happy with all other changes and satisfied that the other comments are addressed.

We understand the concern of the reviewer, and following her/his recommendation, we decided to remove the columns with the flags from the dataset. The new dataset only contains the value of the measured variables together with their uncertainties. We agree that the previous flags are an interpretation of the variability of the data and it is not appropriate to report it with the data.

However, for the purposes of plotting the data in the article, we kept the identified outliers and explain in the text how these outliers were identified. The outliers in the plot are our own interpretation of the data, and does not interfere with other interpretations from data users, who do not see these marked outliers in the final released dataset.

The new version of the dataset can be obtained at the ATTO data portal https://www.attodata.org/ddm/data/Showdata/574 A final doi will be created upon acceptance and we will add it to the final submission version.

We hope that with this change the article can now be accepted for publication.

Best regards,

Carlos A. Sierra, on behalf of all coauthors